# The Exceptional Becomes Everyday: Border Control, Attrition and Exclusion from Within

Regina C. Serpa

Faculty of Social Sciences, University of Stirling, Stirling FK8 4LA, UK; r.c.serpa@stir.ac.uk

**Abstract:** This article examines processes of migration and border control, illustrating the ways by which everyday housing and welfare services function as mechanisms of exclusion in both direct and indirect ways. Using the thesis of crimmigration, the article demonstrates how border controls have become deeply implicated in systems claiming to offer welfare support—and how a global public health emergency has intensified exclusionary processes and normalised restrictive practices. The article compares border controls in two localities—under the UK government's coercive 'hostile environment' policies (based on technologies of surveillance) and a more indirect 'programme of discouragement' in The Netherlands (based on technologies of attrition). The study demonstrates the role of contemporary welfare states in entrenching inequality and social exclusion (from within), arguing that the exceptional circumstances of the COVID-19 pandemic have facilitated the differential everyday treatment of migrants, revealing a hierarchy of human worth through strategies of surveillance and attrition.

**Keywords:** COVID-19; welfare; crimmigration; exclusion; surveillance; attrition



## 1. Introduction: Bordering Practices in the Contemporary Welfare State

A succession of 'crises' observed in the 21st century—including the rise of terrorism, and more recently, the ongoing COVID-19 pandemic—have reinforced the role of borders as defensive barriers against undesirable influences and external threats, helping to construct the contemporary 'problem' of migration (De Genova and Tazzioli 2016). Ostensibly intended to protect national security and promote peace, freedom and prosperity, physical boundaries have served to strengthen societal divisions through intensifying 'paradigmatic borders'—between inside and outside, citizen and noncitizen—in law, public discourse and everyday human interaction (Krasmann 2007; Paasi 2009). Revealed within these societal divisions is a hierarchy of human worth, maintained by the expanding application of state technologies of control, categorisation, surveillance and punishment in migration governance which regards foreigners or noncitizens as suspect persons, and as such, they are assigned a dual identity of 'criminal' and 'migrant' combined: 'the crimmigrant' (Aas 2013, p. 331). Whilst there is no causal link between crime and migration control, a process of 'crimmigration' (Stumpf 2006) has facilitated a new 'state of exception'— where perceived threats to national security and law and order provide the rationale for creating extraordinary measures (Agamben 2005). In the context of support services, such exclusionary practices have been termed 'welfare penalism' and described as a form of 'benevolent violence' (Barker 2012). This article examines how housing and welfare services have become increasingly implicated in decisions relating to border control and national security and how the COVID-19 pandemic has exacerbated the bordering practices that operate in everyday services (Paasi 2009) to incorporate practices of exclusion from within.

This article is structured as follows: first, the methods are outlined in order to demonstrate the development of crimmigration control, in order to provide the theoretical framing of the research, with a focus on the exceptional and everyday, to analyse the role of the state and civil society in migration securitisation and bordered penality. The article examines how migration governance extends into the contemporary welfare state, through

the deployment of technologies of surveillance and attrition. These technologies can be clearly witnessed in the examples of direct and indirect forms of coercion in the UK and The Netherlands, respectively, illustrating the nexus between crimmigration and the use of welfare as a border policing tool. The global pandemic has increased dependency and vulnerability—intensified by processes of attrition (through welfare entitlement) and exclusion from within (via surveillance)—throwing into sharp relief the differential treatment of noncitizens and revealing a hierarchy of human worth.

## 2. Materials and Methods

This study uses a comparative case study approach to consider the different ways of using welfare provision as a border control tool in two superficially contrasting societies: by comparing punitive features of the 'hostile environment' in the UK and the more indirect coercive control mechanisms of the 'programme of discouragement' in The Netherlands. Comparison is useful in highlighting the contingent nature of phenomena, providing insight into the extent to which pre-existing categories are neither natural nor fixed. A comparative method can challenge orthodox thinking, question assumptions and provide theoretical insight (often using interdisciplinary approaches). As Bloemraad (2013) suggested, comparative study provides fertile ground for an analysis of migration processes involving an analysis of a small number of cases to make sense of 'meaningful, complex structures, institutions, collectives and/or configurations of events' (p. 27). There is additional value in studying these issues from a socio-legal perspective as 'international migration implicates rights and legal status as people cross the borders of sovereign nation-states' (Bloemraad 2013, p. 28). Whilst much research on migration has focused on central government policy and the legal process, this article considers how exceptional circumstances are integrated into everyday practices through the operation of welfare and support policies.

The article thus examines the role of coercion and consent in bordering practices and how exceptional practices of exclusion become normalised under what has been termed 'necropolitics' (Mbembe 2003). The study has two central research questions: What is the relationship between welfare, housing and crimmigration control and how important is context (including relatively stable socio-political conditions and exceptional states of crisis) in the configuration of crimmigration control?

The study draws on examples of crimmigration control from two seemingly distinct localities with different ideological underpinnings, separate administrative systems, contrasting public policies and diverse social practices. The UK is selected as a paradigmatic neoliberal regime, based on explicitly coercive and punitive strategies of surveillance to control migration, supplemented by restrictive, highly conditional welfare systems in which crimmigration control is clearly articulated. In contrast, The Netherlands is a regime noted for its social democratic ethos, a facilitative model of social integration and a welfare system based on the principles of inclusion and consent. At the same time, The Netherlands demonstrates emerging features of crimmigration practice that challenge the assumed opposition between neoliberalism and welfarism (Barker 2018). These two examples can therefore provide insight into both causes and effects of arrangements to control, limit and (in some cases) facilitate the settlement of migrant groups. As the study argues, in practice, the two regimes share many assumptions and principles: concerning border control, the creation of 'in' and 'out' groups and exclusionary practices. More specifically, the article argues that indirect strategies of attrition have become a key mechanism of exclusion. As Bloemraad suggested 'you cannot know what is unique, or common, about a particular case unless you have a comparative point of reference' (p. 42).

*Theoretical Framework: Bordering, Crimmigration and Necropolitical Exception*

This research is motivated by three pressing trends in western democratic responses to international displacement and global mobility: (1) the development of observable 'crimmigration' control systems that blur the boundaries between immigration and crim-

inal law, (2) deepening inequality and exclusion based on social divisions such as race, class and gender and (3) increasingly conditional and punitive welfare regimes within an environment of retrenchment and financial austerity. More recently, a fourth trend can be observed during the pandemic, which has ushered in a time of hyper immobility (at least temporarily) as borders close and national as well as localised lockdowns become commonplace, normalising a 'state of exception' in the exercise of unprecedented state power to control contagion—on a global scale. The following section offers an overview of the development of crimmigration control, focussing specifically on the extension of migration governance into the contemporary welfare state.

Globalisation in the last half-century has brought greater interdependence of the world's economies, cultures and populations, accompanied and assisted by technological advances which have enabled growing cross-border flows of investment, people and information (Gundhus and Franko 2016). As the distinction between domestic and international domains is increasingly blurred, 'unwanted' forms of migration (whether humanitarian, undocumented or constituted economic migrants) have become emblematic of a hybrid threat—to national security and sovereignty, on the one hand, and safety and order from *within*, on the other (Koulish and van der Woude 2020). In the US and Europe especially, the responses to the perceived 'threat' of migration have broadly centred on intensifying the 'securitization of migration' (Aas 2013; Guia 2013)—an approach often accompanied by an exclusionary and repressive political and social discourse (Koulish 2010; van der Leun and van der Woude 2013). Such processes of securitisation and exclusion—which radically transform state regulation of migration—have been described as 'crimmigration' to explain the intertwining of criminal and migration control with national security, observed in contemporary western democracy (Stumpf 2006; Guia 2013; Aas 2013).

In this article, bordering practice is conceptualised as involving the exercise of social control by 'inclusionary exclusion' (Agamben 2005), whereby the welfare state apparatus (and other civil society institutions) are co-opted by central authorities in the migration control project (Paasi 2009)—a complicity in crimmigration control which is nevertheless often contested and subject to resistance. However, local sites of 'border resistance' (Weber 2019) tend to be fragmented, ambiguous, idiosyncratic and surpassed by the influences of state control: crimmigration, therefore, has a profound effect on the scope and shape of social welfare *vis-à-vis* bordering practices—transforming humanitarian organisations into 'soft cops of the state' (Poulantzas 1969).

The present study applies Agamben (2005) idea of a 'state of exception' as a dominant paradigm of contemporary government, and the research considers how processes used in a period of crisis (whether financial, social or medical) are instituted within day-to-day social interactions. By combining this perspective with Mbembe (2003) analysis of 'necropolitics' which integrates the 'politics of race' and the 'politics of death' (p. 17), the article investigates how essential services have 'the capacity to define who matters and who does not, who is *disposable* and who is not' (Mbembe 2003, p. 27). From the perspective of 'necropolitical exception' (Farmer 2020), we can therefore explore how welfare structures are co-opted to implement migration control via technologies of surveillance and attrition; processes that exclude noncitizens (often racialised minorities and migrants from the postcolonial Global South) from welfare and housing support (Weber 2019). Through the creation of an exceptional space punctuated by dependence and vulnerability that would otherwise be unacceptable for citizens, welfare becomes a 'necropolitical site of violence' where migrant groups are 'kept alive but in a state of injury' (Mbembe 2003, p. 21) through the conditional delivery and denial of essential services.

By drawing on the literature on crimmigration in the UK and The Netherlands, the next sections contrast the bordering practices deployed in the delivery of accommodation and support services for migrant groups—contrasting the use of 'administrative removals' in the UK (through 'Operation Nexus' and 'Everyone In' policies—based on surveillance and coercion) with a 'programme of discouragement' (based on a principle of consent and attrition) in The Netherlands (van der Leun 2003). Whilst crimmigration can be seen as

a modality of coercion, it should be noted that not all examples of coercion are evidence of crimmigration[1]. Nevertheless, the study shows how these exclusionary technologies of surveillance and attrition have been affected by the COVID-19 pandemic with welfare agencies complicit in policing the border as 'agents of necropolitical exception' (Farmer 2020) in both direct and indirect ways.

## 3. Discussion

This section considers the similarities and differences between the UK and The Netherlands. As discussed above, the two countries were chosen on the basis that they represented contrasting approaches to welfare delivery—on the one hand, a regime dominated by neoliberal ideology (UK) and, on the other, one that has adopted an approach influenced by social democracy (Netherlands) which nonetheless limits the inclusionary nature of the welfare state through hard and soft power to preserve a sense of social security for its members (Barker 2018). Given challenges to assumed opposition of neoliberalism and welfarism, this research suggests that the parallel approaches to the policing of the borders through welfare have become accentuated through strategies of attrition, in an unfolding state of exception during a global health crisis.

### 3.1. UK—Coercion and Technologies of Surveillance

The UK has been extensively criticised for adopting an explicitly punitive approach to migration—for example, by the explicit objective of creating a hostile environment and focusing on immigrant criminality (under Conservative Home Secretaries Theresa May and Priti Patel). Crucially these processes have been extended into welfare policies which have made noncitizens with limited entitlements and precarious legal status increasingly vulnerable to deprivation and homelessness (with rough sleeping used as grounds for removing permission to remain in the UK). As writers such as McKee et al. (2020) have shown, welfare and support agencies (including landlords) have become increasingly recruited in the governance of immigration, using stigma and other forms of power (Tyler 2020) to undermine the legitimacy of claims to migrant rights. These exclusionary processes have been reinforced during the pandemic—as the state of exception (to monitor and limit movement and ensure direct, punitive intervention by the state via information sharing and interagency collaboration) becomes normalised in welfare delivery.

As a consequence of rolling out crimmigration control in the UK since at least 2010 and by enshrining the 'hostile environment' policy in statute within the 2014 and 2016 Immigration Acts, those lacking full citizenship status (particularly those without documented legal status) are increasingly marginalised and excluded from wider society by restricting access to work, welfare and housing. The convergence of criminal and immigration law and its associated exclusionary practices has produced new legal tools available to a range of actors in a variety of institutional contexts, including social welfare providers—amongst others (Bowling and Westenra 2018). Crucially, these social control mechanisms extend far beyond the geographical border to reach deep into civil society, affecting a diverse range of policy areas such as housing, employment, health and education.

Uniquely, within the UK immigration system prior to Brexit, being homeless was the one category into which citizens of countries in the European Union who live in the UK can fall where they are not seen to be exercising their EU member Treaty Rights (as an employee, a jobseeker, a retired person or being economically self-sufficient). The consequence is that, on this basis, a foreign national who ordinarily has the right to live and work in the UK under the European Union's freedom of movement can be subject to administrative removal (deportation) (Serpa 2019). In 2012, 'Operation Nexus'—an interagency collaboration between the police and the Home Office to remove European Economic Area (EEA) nationals without a Right to Reside and/or who have otherwise had encounters with law enforcement—was piloted in London and later rolled out in another six English regions. Between 2012 and 2015, some 3000 'high harm' foreign national offenders (FNOs) were deported under Nexus—many of whom were targeted following

engagement with homelessness and support, rather than criminal justice agencies (Griffiths and Morgan 2017). Deportations enforced under 'Operation Nexus' represent a small but significant part of the deportation machine in the UK which ensnares homeless groups along with (alleged and convicted) criminal offenders, contributing to the deportability of the crimmigrant Other. Based on technologies of surveillance, the UK represents a highly coercive and punitive attitude towards the governance of migration, one which clearly articulates the convergence of criminal and immigration law.

These crimmigration processes have been reinforced through proposals in 2021 (under the Nationality and Borders Bill) including suggestions that migrants should be held in an offshore hub; those arriving without permission could be given prison sentences up to four years (from six months under existing legislation) and those guilty of smuggling migrants could face life sentences (rather than 14 years) (Wadhera 2021). Declaring the asylum system as 'fundamentally broken', Patel has proposed new forms of social control to detect, capture, punish and ultimately banish migrant groups (The Home Office 2021). An explicit connection to crimmigration was demonstrated in Patel's speech in May 2021, criticising local group opposition to deportation and defence of local residents—see, for example, the successful action of local community groups in Glasgow Pollokshields to resist the deportation of two local men (Mackie and Brown 2021). Patel's response was as follows:

> I have a message to those who seek to disrupt the efforts of our enforcement officers. They should think about whether their actions may be preventing murderers, rapists and high harm offenders from being removed from our communities—and they should think long and hard about the victims of these crimes.
>
> (The Home Office 2021)

The severity of the rhetoric towards migrants used by Patel and other would-be enthusiastic crimmigration advocates is mirrored in policy. Responses to COVID-19 have resulted in further mechanisms of social exclusion for migrant groups, revealing differential treatment of noncitizens, reflective of a neocolonial logic constituted by a hierarchy of human worth (Mayblin et al. 2020). On 23rd of March 2020, UK Prime Minister Boris Johnson mandated what he described as the 'very draconian measure' of stopping all 'non-essential contact' with others and putting the country into 'lockdown', telling people in a televised statement they 'must' stay at home (UK Government 2020). Three days later, the UK Government implemented the 'Everyone In' policy and instructed local authorities to invest resources in providing accommodation for people sleeping rough during the pandemic. Crucially, migrants with no recourse to public funds (consisting disproportionately of racialised minorities from the Global South—NRPF Network 2021) and European nationals without a Right to Reside were excluded from the 'Everyone In' policy, a fact which some council leaders and migrant rights advocates challenged in the UK courts. On the 3rd of November 2020, the High Court ruled that councils can provide emergency housing during the pandemic to homeless people who would not normally be eligible for support; however, it was left to individual local authority discretion to use alternative powers and funding to assist those with no recourse to public funds (NRPF) who require shelter and other forms of support (Shelter 2021). While there have been no changes to the policy to impose the NRPF, many cash-strapped local authorities continued to exclude ineligible foreign nationals from 'Everyone In', despite the High Court judgement (NRPF Network 2021).

The exclusion of many foreign nationals from emergency homelessness assistance during the ongoing pandemic continues at a time when new immigration rules come into force, providing the UK Government the power to fully roll out an 'Operation Nexus' style programme of removal across the UK. As of the 1st of January 2021, when the Brexit transition period officially ended, rough sleeping has become grounds for refusal, or cancellation of, permission to remain in the UK. Local authorities across England seem well positioned to accommodate a national roll-out: since early 2019, an increasing number of Home Office agents have been embedded in local authority services to monitor advice and

assistance offered to homeless migrants (Busby 2019). The implications for rough sleepers are considerable—it is estimated that more than a quarter of all street homeless persons in the UK are foreign nationals (Grierson 2020). Despite pressure from human rights groups to end 'Operation Nexus' and put a stop to expanded plans to deport EU rough sleepers across the UK post-Brexit, Home Secretary Priti Patel defended the policy, issuing a Home Office clarification stating, 'permission may only be refused or cancelled where a person has repeatedly refused suitable offers of support and engaged in persistent anti-social behaviour' (Mellor 2021). Charities have warned that the new immigration rules will deter some rough sleepers from seeking help and could push them into modern slavery and other exploitative work (Lister 2020). It is not yet clear how COVID-19 impacts on the law enforcement side of crimmigration policies that harness housing and welfare services to facilitate deportations; however, it is apparent that homeless foreign nationals—as the only group of rough sleepers excluded from the 'Everyone In' policy—have become much more visible and therefore easily identifiable as candidates for removal.

The effect of such technologies of surveillance and attrition, therefore, is the entrenchment of the criminalisation of migration in the UK by combining civil exclusions (relating to restricting access to homelessness support services) with deportation as an adjunct to criminal penalty (lacking settled status now constituting an illegal stay for EEA nationals in the UK). Deploying interventions based on force and control—and supported by the identification, categorisation and surveillance functions of the welfare state—deportation secures compliance with immigration policy by removing the possibility of choosing not to comply. By excluding many groups of migrants from the 'Everyone In' policy to remove homeless persons from the street, the pandemic (in combination with Brexit) lays the groundwork to intensify and expand such exceptional use of force to deport unwanted foreign nationals. This example not only illustrates how welfare providers have been made complicit by policy in migration control in a UK context but also how such imagining of 'immigrant criminality' is vital to understand the perceived political expediency of instituting a hostile environment for migrants and in general the legitimacy of social exclusion in societies (Franko 2019). In order to provide a contrasting approach, the next section considers how The Netherlands has approached the governance of migration in housing and welfare delivery.

### 3.2. Technologies of Attrition: The 'Programme of Discouragement' in The Netherlands

The Netherlands has been long commended for adopting a tolerant and humane approach in the treatment of migrants (Van der Woude et al. 2014). However, since the 1990s, Dutch immigration policies have been characterised by restrictive admission policies, increased exclusion of unauthorised migrants and greater pressure for migration control (Engbersen et al. 2006). Increasingly, the trend towards the securitisation and criminalisation of migrants observed in neoliberal regimes (such as the UK and the US) is emerging in The Netherlands and elsewhere in northern welfare states, prompting crimmigration scholars to question the assumed opposition between neoliberalism and welfarism and scrutinise the exclusionary nature of the welfare state's inclusionary logic (Barker 2018; Franko 2019).

Since the 1990s, The Netherlands was among the first countries in the European Union to reform immigration policy specifically targeting irregular migration, set within a context of a (financially and ideologically) pressurised welfare state—diminishing border controls, despite the EU principle of Freedom of Movement (van der Leun 2003; Leerkes et al. 2012). Policies of attrition have been implemented to prevent entry, exclude from social benefits and public assistance and expel irregular immigrants (van der Leun 2006)—this, in combination with increasingly managerialised austere state-run services, has resulted in growing desperation amongst 'unauthorsied' migrant groups. Alongside tightening migration controls, immigration-related penalties (such as deportation) have been introduced for criminal offences—reversing a trend towards limiting penal power in the turn towards crimmigration control. According to Van der Woude et al. (2014), over the 21st century,

a 'humane paternalism', historically characteristic of the Dutch criminal justice system, has gradually been replaced with a process of 'managerial instrumentalism', deploying punishment as a 'cultural agent'. This newfound drive towards penalty signalled that 'the Dutch have purged themselves of the misplaced leniency of the past and are no longer afraid to punish' (Downes and Van Swaaningen 2007, p. 66).

Moreover, an association between ethnicity and social problems (especially crime and disorder perceived to be linked with migration from Morocco and the Antilles—Van der Woude et al. 2014) has gained political traction, with broad public support for stricter measures in The Netherlands commonly practiced elsewhere in Europe, such as deportation of immigrants who had committed crimes and 'soft' deportations (Versteegh and Maussen 2012). The punitive turn taken in The Netherlands, amid a backdrop of continuously hardening political and social discourse on immigration (and immigrants), can be traced to the 1980s with the Ministry of Justice white papers *Crime and Society* and *Law in Motion*. These policies have been generally regarded as a turning point in Dutch criminal justice policy, forging an indelible link between concerns about immigration and integration, on the one hand, and crime and safety, on the other, in popular and political imagination (Van der Woude et al. 2014). The sharper end of the crimmigration control system in The Netherlands can be observed in the growing criminalisation of migration (for example, the creation of specific immigration-crime offences including criminalising illegal stays). This 'immigrationalization of criminal law' (Legomsky 2007) includes deportation as an adjunct to criminal penalty, as well as expanding grounds for administrative detention on the basis of an immigrant's criminal background. It is important to note that, although deportation is an 'adjunct to criminal penalty', it is not considered a form of punishment despite having clear punitive consequences for the expelled (Van der Woude et al. 2014).

In response to public perception of the connection between immigration and social disorder, the Dutch government has turned to increased state coercion (Barker 2012). For example, governmental and quasi-governmental services (including welfare departments and social housing providers) are obliged under the 1998 Linking Act to conduct residency checks prior to giving access to certain services and benefits (Leerkes et al. 2012). Although new investigative powers of public sector and non-profit intermediaries have been expanded by the 2000 Aliens Act (amended in 2013 by the Modern Migration Policy and National Visa Acts), the amount of active surveillance of unauthorised migrants performed by civil service organisations in The Netherlands is limited (van der Leun 2003), owing in part to the resistance of relief organisations in policing migration. Rather than embark on a programme to fully converge criminal and immigration control (as is the aim of British migration policies), it would appear the Dutch approach to migration governance is less punitive and more permissive but nonetheless coercive in its exercise of social control and attrition. Given the emerging nature of the law enforcement side of crimmigration policies in The Netherlands, crimmigration is taken as a 'sensitizing concept' (Van der Woude et al. 2014) in understanding the ways in which welfare is used as a border policing tool.

Amid this backdrop of increasing criminalisation of migration, Dutch civil society includes a paradoxical merger of humanitarian care and securitisation imperatives (Kox and Staring 2020). In this context, humanitarian organisations refer to the wide range of agencies that provide relief to alleviate the hardship of migrants—critically, as Kox and Staring (2020) argued, these support organisations do so whilst simultaneously 'reproducing the causes of migrants' suffering and legitimizing restrictive migration policies' (p. 3). As van der Leun and Bouter (2015) demonstrated, internal border controls have rendered migrants without legal status wholly dependent on the (material and non-material) support of humanitarian organisations. Originally established to offer support to groups excluded from state-provided forms of support—many of these agencies emerged from protest movements against restrictive migration policies—these 'emergency relief' organisations now find themselves in the 'ambiguous' position of advocating for migrant rights whilst collaborating with central authorities in migration control (Kox and Staring 2020, p. 3). Whilst there is a measure of organisational resistance to these processes, empirical evidence

suggests that unauthorised migrants largely consider these humanitarian organisations to be part and parcel of the Dutch migration control system (Kox and Staring 2020).

Thus, since the 1990s, humanitarian organisations have worked in close collaboration with their respective municipalities, limiting their power to resist central migration policies (Kalir and Wissink 2016). In the Dutch case, such organisations are, in effect, coerced by local municipalities via 'control by compliance' (Baines and van den Broek 2017) into implementing a programme of 'soft deportations' (called Assisted Voluntary Return), which function in addition to (or as a replacement of) state deportations (Leerkes et al. 2012). At the same time, participation in 'migration policing networks' has been contested by multiple acts of 'micro-refusal,' posing a challenge to state-centric bordering practices (Weber 2019; King 2016). Such local resistance to harsh immigration policies was articulated in April 2015 when several Dutch municipalities issued a statement in the daily newspaper *Volkskrant* refusing to cooperate with a decision by the then Dutch cabinet to refuse temporary shelter to failed asylum seekers and instead confirmed their continued commitment to provide '*bed-bad-brood*' arrangements providing shelter, bathing facilities and food relief for unauthorised migrants, rather than 'put or leave rejected asylum seekers out on the street' (Versteegh 2016, p. 366).

Even though central authorities have broadly opposed support for migrants residing in The Netherlands unlawfully, the Dutch government has never signalled an intent to criminalise such support—although there are financial consequences for municipalities that fail to meet state's expectations concerning resource management and policy delivery (Gerard and Weber 2019). Such strategies of attrition mean that, in exchange for support from local municipalities, humanitarian organisations can generally only assist those migrants who meet pre-determined eligibility criteria. The consequence is that only migrants who have a case for legal residency or who agree to voluntarily return to their country of origin are offered support. Emergency relief organisations dependent on municipality support are therefore effectively forced to exclude migrant clients falling outside these criteria (LOS Foundation 2014), demonstrating how indirect control is exercised through systematic withdrawal of services, rather than active intervention. The result is the creation of a 'structurally embedded border' involving 'migration policing networks' (Weber 2013) recruited to have both direct and indirect bordering effects. Denials of service thus create metaphorical but nevertheless powerful borders, leading to differential forms of social, political and economic in/exclusion.

Over the course of the pandemic, this attritional 'control by compliance' has involved 'cutbacks coercion' where state control is exercised through the failure to fund services to adequate levels. In contrast to Hall (2004) definition of coercion, such 'thwarted rights and stunted care' suggest that neglect originates at a systemic level, rather than being arbitrarily imposed, as organisations are under pressure to (reluctantly) act as conduits of control of scarce resources (Baines and van den Broek 2017, p. 142). Such attritional strategies are witnessed in the aforementioned closure and consolidation of several bed-bath-bread facilities over the course of the pandemic, resulting in increased numbers of homeless asylum seekers in desperate need of emergency relief (de Waard 2020). In The Netherlands, where unauthorised migrants are excluded from all formal markets and welfare arrangements (and only allowed essential healthcare, legal aid and primary and secondary education), studies on irregular migration have shown that it has become increasingly difficult to survive without a Dutch residence permit (Burgers and Engbersen 1999; Engbersen et al. 2002; Staring and Aarts 2010). In European states with strong welfare safety nets, such as in Scandinavian countries, the principles of universalism and inclusivity can be sustained, despite treating non-nationals punitively—simply because the 'crimmigrant Other' falls outside the responsibility of the welfare state (Barker 2012; Gundhus 2020). However, the closure of core facilities has intensified these struggles, resulting in deep social exclusion with many becoming dependent on (informal and formal) forms of support, and those lacking resources become vulnerable to exploitation and the possibility of engaging in survival crime (Van der Woude et al. 2014). In this way, the

crimmigration–welfare nexus is sustained through an association between migration and extra-legal activities.

## 4. Conclusions

Despite differences in ideology, emphasis, institutional support and administrative approaches, the article highlights the similarities and differences in approaches to border control in the UK and The Netherlands. Whilst the UK adopts many features of a classic coercive state (dominated by central- and local-level state institutions and governed by technologies of surveillance), the Dutch approach is characterised by indirect coercion (administered increasingly by 'humanitarian' organisations), although underpinned by technologies of attrition. However, the underlying pressures—to reduce resources, limit immigration, control the behaviour of migrant groups, criminalise certain activities and use the agencies of state and civil society to reinforce stigma and social exclusion—are increasingly dominant in the design of welfare systems reflecting a hierarchy of human worth. Indirect coercion has become more apparent in times of crisis, with the effect of increasing dependency and vulnerability simultaneously; technologies of attrition that systematically deny noncitizens access to housing and welfare have therefore become an effective mechanism of exclusion from everyday services, and by extension, quotidian life.

The process of crimmigration implicates housing, welfare systems and other facets of civil society (including educational and healthcare settings) in everyday policing of migration. The use of crimmigration as explicit coercion (in the UK) and as a 'sensitising concept' (in The Netherlands) has potentially severe consequences for noncitizen groups (particularly for those unable to document legal status). The retrenchment of civil and social rights accompanying the extension of crimmigration control across multiple domains of social life (namely, with the introduction of accessorial liability in welfare provision and creating civil exclusions across a range of institutional contexts) has directly contributed to the growing economic and social precarity of migrant groups. For some migrants, the interaction of several systems, such as immigration, labour, welfare and housing markets, creates a reinforcing cycle of poverty that, once trapped, is difficult to escape (Dwyer et al. 2018). Found in a 'Catch-22' situation, socially excluded migrants become unable to afford housing due to low pay or no income, which in itself is a barrier to securing employment (for example, due to costs of travel) necessary to pay for accommodation (Maycock and Sheridan 2012). Migrants facing work and welfare restrictions due to their immigration status have few housing options and in extreme cases can result in homelessness and destitution (Dwyer et al. 2018; Edgar et al. 2004; Fitzpatrick et al. 2013). Engagement in 'survival crime' enhances a post-crimmigration nexus, a process that can legitimate further coercive measures. The COVID-19 pandemic has added to the desperate situation many migrant groups face, as many western democracies respond to an increase in asylum claims by (temporarily) suspending asylum protections, closing emergency relief and shelter provision and, in some instances, extending detention periods leading to the overcrowding of vulnerable adults and children in unsafe and inhumane conditions (Migration Data Portal 2021; Aal et al. 2021). The consequence of these processes is that a strategy of attrition has become more profound, leading to hierarchies of human worth as migrant groups are denied access to core services.

As a mode of social control in the welfare state context, contemporary bordering practices have served to reinforce marginalisation, dependency and destitution—processes that have intensified under a protracted state of exception which has resulted in increased use of indirect strategies of attrition, rather than direct controls through surveillance and explicit coercion. Notwithstanding these processes that render welfare providers complicit in crimmigration control through policy, the case studies presented here also demonstrate a measure of resistance reflected in the legal challenges brought against the UK Government's 'Everyone In' policy and the refusal of relief organisations and municipalities to deny essential services to unauthorised migrants in The Netherlands. Similarly, other research studies have signalled the potential of local-level, multiple, small-scale, temporary but

significant strategies to facilitate the emancipatory potential of services through resistance (Weber 2019; King 2016), representing opportunities to create inclusive settlements from within.

**Funding:** This research was funded by the ESRC, grant number ES/V01210X/1.

**Acknowledgments:** I want to acknowledge Kim McKee of the University of Stirling, Maartje van der Woude of Leiden University, and the three anonymous referees for their invaluable feedback on this research.

**Conflicts of Interest:** The author declares no conflict of interest.

## Note

1    I am grateful to one of the anonymous referees for making this point.

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
