# Peer review of "The Exceptional Becomes Everyday: Border Control, Attrition and Exclusion from Within"

_socsci, doi:10.3390/socsci10090329_

Round 1

Reviewer 1 Report

This paper is an insightful contribution that comparatively explores how welfare services are used to boost the crimmigration agenda in the Netherlands and the UK. It is based on a detailed description of bordering processes in the welfare field in both countries. In exploring this frequently overlooked dimension of crimmigration policies, it sheds much needed light on how the welfare system treats undocumented noncitizens and how it contributes to their differential inclusion.

Whilst the paper presents a robust exploration of the topics under study, its analysis of the impact of the coronavirus pandemic on this field remains inconclusive and is somehow unconvincing. The paper claims that ‘the Covid-19 pandemic (…) has intensified crimmigration control, by polarising bordering practices at either end of the control-coercion continuum of care.’ (p. 4, 164-166), whereas is far for evident that this has been the case. The paper describes in detail how the pandemic has strengthened the political debate regarding rough sleeping noncitizens in the UK (pp. 4-6) and points out relatively similar developments in the Netherlands (p. 8). However, the paper does not explore if and how the pandemic has affected the law enforcement side of crimmigration policies. It incidentally points out that ‘many western democracies’ are ‘detaining a growing population of vulnerable adults and children in unsafe and inhumane conditions’. In stark contrast to this claim, it is known that the coronavirus altered immigration detention practices in Europe by leading many western European jurisdictions to significantly curtail the number of detainees – especially in the first wave of the pandemic. Conditioned by anti-coronavirus restrictions, deportation policies have followed the same trend. The number of deportations enforced in the EU declined by 39.4 per cent from 2019 to 2020 (source: Eurostat). Eurostat does not provide data on recent UK deportation practices. However, it is evident that Britain is no exception to this general trend. In fact, deportation measures were already on the decline in the UK; the number of deportations annually enforced dwindled by 55.2 per cent from 2015 to 2019 (source: Home Office). In sum, the paper should further examine if and to what extent the covid pandemic has actually intensified crimmigration practices in Europe. In so doing, it may reconsider the relevance of Giorgio Agamben’s controversial thesis of the state of exception for this study. As it currently stands, the reference to Agamben’s thesis is incidental, vague, and apparently dispensable.

An additional aspect to be rethought has to do with the description and analysis of specific law enforcement practices in both the Netherlands and Britain. In the UK case, the paper does not clearly differentiate between crimmigration measures targeting rough sleepers and unhoused people and removal practices targeting formerly sentenced (‘high crime’) noncitizens (pp. 4-5), whilst they are very different phenomena from both legal and political viewpoints. In the Dutch case, the paper apparently emphasises the relevance of two crimmigration measures which are not quintessential traits of the Dutch crimmigration apparatus, but measures that are widely implemented in the majority – but not all – of European jurisdictions: a) the criminalisation of immigration offences (p. 7, 310-313); b) ‘soft deportations’ (p. 7, 351-352). Curiously, the Home Office reports that the majority of removals enforced in the UK are actually ‘soft (voluntary) deportations’ – around 60-70 per cent over the last half of the 2010s. The paper may want to reconsider whether these specific measures are particularly suitable to describe crimmigration processes in these two jurisdictions.

In addition to these two vital aspects, there are some other minor points which require further attention:

  1. The paper may want to consider Vanessa Barker’s recent work on the role played by national welfare for state control purposes, especially its 2018 book Nordic Nationalism and the Penal Order;
  2. The paper may further elaborate the significance of Foucault’s governmentality perspective for its analysis (p. 3, 97-99), because welfare-based forms of control are largely manifestations of state coercion, not control practices carried out by non-state actors;
  3. The paper incidentally mentions ‘EEA nationals’ (p. 4), a notion that is hardly used outside of the UK. A definition of this acronym might help international readers to understand this concept;
  4. The paper mentions a High Court ruling issued on ‘the 3rd of November 2021’ (p. 5, 219-220).

Author Response

Response to Reviewers

I would like to thank all three reviewers for their insightful and constructive comments. The paper has now been significantly rewritten (showing track changes). The information below explains how the main comments raised by the reviewers have been addressed in the amended paper.

  1. Analysis of coronavirus is unconclusive and unconvincing (reviewer 1)
    1. The article has been rewritten to demonstrate how crimmigration control infiltrates housing and welfare services - via technologies of surveillance and attrition. The argument is that this has produced the effect of social exclusion from within and facilitated banishment.
    2. I have emphasised how the crisis of Covid-19 has intensified attrition and exclusion – throwing into sharp relief the differential treatment of migrants and revealing a hierarchy of human worth (through the idea of ‘necropower’).
  2. Provide clearer description and analysis of specific law enforcement practices in both the Netherlands and Britain (reviewer 1 and 2). Consider migratory pressures in each country (reviewer 3).
    1. I have reworked the paper to emphasise the context in the two countries to highlight both similarities and differences with a focus on hostile environment policies (in the UK – focusing on a coercive and punitive programme) and the ‘programme of discouragement’ (in the Netherlands – based on indirect coercion practices).
  3. Article is over-ambitious and involves too many disparate theories – should focus on core concepts (reviewer 2). Theoretical framework should be in separate paragraph (reviewer 3).
    1. I have reframed the theoretical perspective to focus on a ‘necropolitical exception’ perspective which focuses on technologies of surveillance (migration policing networks) and attrition (removing/prohibiting essential services for migrants).
    2. I have adapted the Agembian perspective (retaining the argument of the exception becomes the everyday), but included a focus on ‘necropower’ which clearly articulates a hierarchy of human worth and the desire to ‘keep alive but in a state of injury’ (through technologies of attrition, which exclude from within). This latter perspective has become more apparent during Covid).
    3. I have applied a comparative case study method - to contrast the approach in the UK (deploying direct control through administrative removals) with the Netherlands (using indirect mechanisms such as ‘soft’ deportations and encouraging voluntary returns). The effect of Covid has been to emphasise indirect efforts to exclude via attrition rather than direct efforts to banish.
  4. Loose application of crimmigration - used interchangeably with the broader concept of the securitisation of borders (reviewer 2). Provide clearer definition of crimmigration (reviewer 3)
    1. I have provided a clear definition of crimmigration and included a more explicit demonstration as to how this concept has been applied within housing and welfare systems - in a literal sense in the UK and the extent to which it is resisted in the Netherlands.
  5. Reconsider research questions and data (reviewer 2). I have reworded the research questions as follows:
    1. RQ1 – how are housing and welfare systems used to deliver crimmigration?
    2. RQ2 - how important is (socio-political) crisis in achieving crimmigration control?
    3. Not using empirical data but developing a framework in which to understand processes of exclusion by embedding an analysis of necropolitical exception within a crimmigration perspective
  6. Core arguments need to be clearer (reviewer 2)
    1. The core argument is that crimmigration is a useful framework to explain border control through technologies of surveillance and attrition. These technologies can be clearly witnessed in direct and indirect forms of coercion (in the UK and the Netherlands, respectively). The global pandemic has intensified both attrition (through welfare entitlement) and exclusion – throwing into sharp relief the differential treatment of migrants and revealing a hierarchy of human worth (through the idea of ‘necropower’).
  7. Results to be renamed as discussion (reviewer 2)
    1. I have amended the paper to incorporate this change.
  8. Need to cite additional sources (reviewer 2)
    1. I have added core references including the seminal work of Stumpf (2006), as well as Mayblin et al (2020) and Weber (2019).
  9. Conclusions are very broad and need to cite specific practices (reviewer 2). Focus discussion much more narrowly on the nexus between crimmigration and the use of welfare as a border policing tool (reviewer 2)
    1. Necropolitical exception helps explain how direct and indirect forms of crimmigration are manifested in different welfare contexts. Operation Nexus in the provides an example of punitive coercion (in the UK) and the ‘programme of discouragement’ (in the Netherlands) illustrates how indirect controls operate (based on a model of consent). Indirect coercion has become more apparent in times of crisis, with the effect of increasing dependency and vulnerability simultaneously; technologies of attrition have therefore become an effective mechanism of exclusion from everyday services.

Thank you for the opportunity to address reviewers’ comments.

Sincerely,

Authors

Reviewer 2 Report

There is a growing interest in the use of welfare as a form of border control – and also an interest in the treatment of non-citizens during the pandemic. This article spans those two themes, as well as using a crimmigration framework as invited by the special issue, and should therefore find a readership.

My assessment is that the article is trying to do way too much at present and does not fully establish many of the claims made, but might do so with some concerted refocusing onto the core themes of welfare policing and crimmigration.

Contextualisation/theoretical framing

The securitisation of migration is well covered in the theoretical framing section, but I feel there is quite a loose application of the concept of crimmigration which seems to be used interchangeably with the broader concept of the securitisation of borders.

Additional theoretical concepts referred to are governmentality, the control-coercion continuum, stigma, crisis, state of exception, the citizen/denizen contrast, the right to have rights (although Arendt is not cited), and inclusionary exclusion (attributed to Gundhuus, but not the earlier articulation of exclusionary inclusion by Agamban).

Unless all of this complexity can be brought together in an integrated theoretical framework, which would be quite a task, I believe it would be better to focus on just those core concepts that are going to be carried through the analysis. I thought the control-coercion continuum might be a possible contender.

Research methods

The author adopts a comparative socio-legal approach comparing the UK and Netherlands. These sites have been chosen, inter alia, because they reflect neoliberal and social democratic countries within Europe. As the author points out, there is considerable existing scholarship (e.g by Vanessa Barker) that challenges the idea that social democracies necessarily adopt more inclusive border control policies. Nevertheless, a close comparison of the two could be fruitful.  

I think we need to know precisely what questions are being addressed and what data is being brought to bear on those questions, so that the two countries can be compared systematically. It seems to me that setting out to compare ‘the governance of migration’ in these two locations is just too non-specific as it spans migrant groups with very different legal statuses, and a potentially very wide range of policies.

At the moment we have two different stories being told in two jurisdictions and it is difficult to see how to compare and contrast them. This is why I am wondering whether the control-coercion continuum might be a useful analytical device, so that the two countries might be placed at different or similar points on the continuum across a number of themes.

Alternatively – or possibly in addition - a case study approach might be suitable, where the focus of each case study could be different e.g. Everyone In/Operation Nexus in the UK, ‘soft deportations’ in the Netherlands. The rationale for proceeding this way might be that they illustrate different ways of using welfare provision as a border control tool.

Quality of arguments

I think at present the core arguments are not clearly made out because the discussion is rather too broad, and many statements are not fully supported. 

It was not always clear to me that the material presented were examples of crimmigration. On other occasions, examples that did align with the crimmigration thesis did not seem to fit the focus on welfare provision. I think successfully integrating these two themes into the analytical framework is vital for the arguments to succeed as intended. I’ll illustrate these issues using some examples from the UK section.

For example, we read that: ‘The process of crimmigration implicates housing, welfare systems and other facets of civil society (including educational and healthcare settings), in everyday policing of migration.’ But the article at present does not really establish what this means. Readers may well accept that these processes operate as powerful forms of border control, but in what way do they reflect the convergence of immigration and criminal law and process?

Most of the examples given involve administrative removal. Operation Nexus seems more reflective of crimmigration as it involves both police and Home Office personnel and, importantly, is linked in this passage to welfare provision for the homeless.

“The effect of the policy therefore is to entrench the criminalisation of migration in the UK, by combining civil exclusions (relating to restricting access to homelessness support services) with deportation as an adjunct to criminal penalty (lacking settled status now constituting an illegal stay for EEA nationals in the UK).”

Focusing on these arguments seems to be the most promising way forward.

Another explicit link to crimmigration in the UK section comes from a statement by Home Secretary Priti Patel, but that reflects political discourse, not the elucidation of actual processes on the ground.

"I have a message to those who seek to disrupt the efforts of our enforcement officers. They should think about whether their actions may be preventing murderers, rapists and high harm offenders from being removed from our communities - and they should think long and hard about the victims of these crimes (Home Office, 2021)".

The distinction between discourse and praxis has of course formed one of the key debates within crimmigration literature, but the author needs to locate this statement within that debate about what crimmigration actually means.

Elsewhere we read:

"By excluding many groups of migrants from the ‘Everyone In’ policy to get homeless persons off the street, the pandemic (in combination with Brexit) lays the groundwork to intensify and expand such exceptional use of force to deport unwanted foreign nationals. This example not only illustrates how welfare providers have been complicit in migration control in a UK context … "

But no data has been produced to establish how welfare providers have been complicit and the conclusion seems to confuse policy with practice. In previous decades, for example, local councils resisted exclusionary policies through legal channels. Perhaps welfare providers have been ‘made complicit’ through policy, but this distinction needs to be carefully maintained.

The idea of continuum is also applied briefly within the analysis: ’The idea of a continuum, rather than strict dichotomy between inclusion and exclusion, demonstrates the level of convergence in the two countries.’ But again, I don’t see that this has been demonstrated empirically, and I am not sure that it is coherent to argue that a continuum promotes convergence. 

Presentation of results

I suggest that the ‘results’ section might be better named a ‘discussion’ since it does not really present empirical findings as generally understood.

Additional references

It seems unusual not to cite Juliet Stumpf’s original formulation of the crimmigration thesis or the debates that have followed.

Stumpf, J. P. (2006). "The Crimmigration Crisis: Immigrants, Crime, & Sovereign Power." American University Law Review 56(2): 356-420. 

There is also some emerging work on policing borders through welfare that could be consulted eg.

Mayblin, L., Wake, M. and M. Kazemi. (2019). "Necropolitics and the Slow Violence of the Everyday: Asylum Seeker Welfare in the Postcolonial Present." Sociology.

Weber, L (2019) ‘From state-centric to transversal borders: Resisting the ‘structurally embedded border’ in Australia’, Theoretical Criminology 23(2), p228-246

The large literature on penal humanitarianism might be relevant also.

Conclusion

The article reaches a very broad conclusion: ‘Despite differences in ideology, emphasis, institutional support and administrative approaches, the article highlights how the governance of migration has taken similar trajectories in both the UK and the Netherlands’. This seems to undervalue the work that has been done to tease out the specifics – which is an essential element of comparative work. I think the broad focus on ‘the governance of migration’ rather than the policing of borders through welfare more specifically, may be the main culprit.

Another concluding observation, highlighted in the abstract, seems to shift the focus away from the central problematic of the inclusion of non-citizens: ‘In contrast, a fair and inclusionary approach can facilitate the emancipatory potential of services; ones based on the principles of inclusion and citizenship’ (emphasis added).

Overall, my suggestion is to try to focus this discussion much more narrowly on the nexus between crimmigration and the use of welfare as a border policing tool in order to achieve more depth. 

Author Response

(The authors gave the same response as above.)

Reviewer 3 Report

Dear authors,

I would suggest to include ‘Theoretical framework’ in a separate paragraph, and not a subparagraph as shown in the text. (Eg. 3. Theoretical framework).

I would elaborate a bit more the concept of crimmigration, so the readers fully understand the concept. We may suppose that people reading this paper have no clue on the concept, and they need more context. Indeed, I would suggest to broaden your theoretical framework.

When you compare the UK and the Netherlands you should take into account migratory pressure in each country. If we consider that the UK receives a more intense pressure it would also explain the measures taken, in contrast to the Netherlands. I mean this is a structural matter, and we may consider many dimensions. English language, learned in almost every country, would be a useful tool which may explain the ‘desire’ to try a better life for migrants in English speaking countries. This fact may not be the same if we analyze the case of the Netherlands.

I find you should reconsider your perspective and offer a more extensive analysis

Author Response

(The authors gave the same response as above.)

Round 2

Reviewer 2 Report

There has clearly been a significant effort made to address the previous feedback. 

Bringing the discussion under the umbrella of necropolitics has helped to 'tidy' up the theoretical framing, and is consistent with the idea of attrition as a technology of border governance.

There seems to be more consistency now across the associated ideas of border securitisation, exception and hierarchies of humanity.

My reaction is still that crimmigration is used in places as if it equates to coercion. I would say crimmigration is a specific modality of coercion, but not all coercion within border control is an example of crimmigration.

The influence of crimmigration is apparent in most of the UK policies discussed, esp. Op Nexus, but less evident in the Netherlands.

It is now clearer that this is a key part of the author's argument, and helpful explanations are added about the relevance of crimmigration in the Netherlands section by referring to crimmigration as a 'sensitizing concept' and using the idea of 'survival crime' to posit a crimmigration-welfare nexus. 

I would urge the author to revisit this early reference to crimmigration: "This entanglement of crime and migration control, a process described as ‘crimmigration’ (Stumpf, 2006), has instituted a near permanent ‘state of exception’ ..."  This seems to assert a causal link that I am not sure is intended.

I would also encourage the author to consider omitting the final reference to 'transversal borders' in the conclusion, since this has not been discussed in the article so comes somewhat 'out of the blue'.

Overall, I think the arguments are now much clearer. Thank you for the opportunity to read this article again. 

Author Response

Response to second reviews

Many thanks for the helpful and speedy responses to the amended draft. I have now made changes as suggested by reviewers. These include:

  • reference to crimmigration as a specific modality of coercion and acknowledgement that not all coercive practices are examples of crimmigration;
  • making a clearer distinction between the UK (dominated by coercive practices) and the Netherlands (where crimmigration is used more as a ‘sensitising concept’;
  • a reference to survival crime as part of a post-crimmigration nexus in the conclusion
  • I have removed the reference in the final sentence to ‘transversal borders’.

I am grateful for the very helpful feedback, the close attention given to my work (by editors and referees) and hope that the article can now be accepted for publication.

Reviewer 3 Report

The author took into consideration the advices from the reviewer. I find that this work is now appropriate for publishing.

Author Response

(The authors gave the same response as above.)
